# Melatonin Ameliorates Diquat-Induced Testicular Toxicity via Reducing Oxidative Stress, Inhibiting Apoptosis, and Maintaining the Integrity of Blood-Testis Barrier in Mice

**DOI:** 10.3390/toxics11020160

**Published:** 2023-02-08

**Authors:** Li Yang, Jianyong Cheng, Dejun Xu, Zelin Zhang, Rongmao Hua, Huali Chen, Jiaxin Duan, Xiaoya Li, Qingwang Li

**Affiliations:** 1College of Animal Science and Technology, Northwest A&F University, Yangling, Xianyang 712100, China; 2Chongqing Key Laboratory of Herbivore Science, College of Animal Science and Technology, Southwest University, Chongqing 400715, China; 3College of Pharmacy, Shenzhen Technology University, Shenzhen 518000, China; 4School of Life Science and Engineering, Southwest University of Science and Technology, Mianyang 621000, China; 5College of Animal Science, Shanxi Agricultural University, Taiyuan 030801, China

**Keywords:** testicular toxicity, melatonin, diquat, oxidative stress, apoptosis, blood-testis barrier

## Abstract

Diquat is a fast, potent, and widely used bipyridine herbicide in agriculture and it induces oxidative stress in several animal models. However, its genotoxic effects on the male reproductive system remain unclear. Melatonin is an effective free-radical scavenger, which has antioxidant and anti-apoptotic properties and can protect the testes against oxidative damage. This study aimed to investigate the therapeutic effects of melatonin on diquat-induced testicular injury in mice. The results showed melatonin treatment alleviated diquat-induced testicular injury, including inhibited spermatogenesis, increased sperm malformations, declined testosterone level and decreased fertility. Specifically, melatonin therapy countered diquat-induced oxidative stress by increasing production of the antioxidant enzymes GPX1 and SOD1. Melatonin treatment also attenuated diquat-induced spermatogonia apoptosis in vivo and in vitro by modulating the expression of apoptosis-related proteins, including P53, Cleaved-Caspase3, and Bax/Bcl2. Moreover, melatonin restored the blood-testicular barrier by promoting the expression of Sertoli cell junction proteins and maintaining the ordered distribution of ZO-1. These findings indicate that melatonin protects the testes against diquat-induced damage by reducing oxidative stress, inhibiting apoptosis, and maintaining the integrity of the blood–testis barrier in mice. This study provides a theoretical basis for further research to protect male reproductive health from agricultural pesticides.

## 1. Introduction

Diquat (1,1′-ethylene-2,2′-dipyridylium), a non-selective bipyridine herbicide, has been widely used in agricultural production as a paraquat substitute. However, diquat residues in food or accidental ingestion of diquat are extremely harmful to humans. In addition, clinical studies have shown that approximately one-third of diquat-related acute illnesses are work-related [1] and that diquat is toxic to various organs, including the liver, heart, and nervous system [2]. In particular, diquat is toxic to the kidneys of both humans and animals, causing acute kidney injury [3,4,5] and renal failure [6]. Research involving adult male zebra finch (Taeniopygia guttata) showed that diquat treatment can reduce sperm velocity and shorten the sperm midpiece [7]. In mammals, chronic exposure to diquat decreases ovarian weight and induces oxidative stress, granulosa cell apoptosis, and poor developmental potency of oocytes [8]. However, data on male reproductive toxicity of diquat in mammals are limited. 

Melatonin (N-acetyl-5-methoxytryptamine), an indoleamine synthesized primarily by the pineal gland, modulates multiple physiological processes in mammals [9,10,11]. Notably, the role of melatonin in male fertility has been long established. Melatonin modulates testicular functions by regulating testosterone production in Leydig cells [12], proliferation, and energy metabolism in Sertoli cells [13], and consequently influences spermatogenesis [14]. Specifically, melatonin and its metabolites are potent free-radical scavengers that protect the testes against oxidative damage caused by hyperthermia [15], irradiation [16,17], and environmental toxins [18,19]. However, whether melatonin can alleviate oxidative stress and testicular damage induced by diquat remains to be elucidated.

In the present study, a 28 day-long treatment with diquat pathologically and morphologically damaged the seminiferous tubules and inhibited the growth of mouse testes. Moreover, melatonin significantly prevented diquat-induced dysfunction of testes by inhibiting reactive oxygen species (ROS) production and apoptosis, and maintaining the integrity of the blood-testis barrier (BTB). Our study provides novel insights into the toxicological effect of diquat in male reproductive biology and the role of melatonin in countering diquat injuries in the testes. These results will help facilitate future studies on human reproductive health against pesticides used in agricultural practices.

## 2. Materials and Methods

### 2.1. Chemical and Reagents

Diquat (Cat#: 6385-62-2) and melatonin (Cat#: 73-31-4) were purchased from Sigma-Aldrich (St. Louis, MO, USA). The antibodies anti-Sod1 (Cat#: WL01846), anti-Gpx1 (Cat#: WL02497a), anti-Bax (Cat#: WL01637), anti-Bcl2 (Cat#: WL01556), anti-Cleaved Caspase3 (Cat#: WL01992), anti-P53 (Cat#: WL01919), anti-ZO-1 (Cat#: WL03419), anti-Occludin (Cat#: WL01996), anti-β-catenin (Cat#: WL0962a), and anti-Connexin43 (Cat#: WL02837) were purchased from Wanleibio (Shenyang, China), while anti-Tubulin (Cat#: AF1216) was obtained from Beyotime (Shanghai, China). Unless otherwise indicated, the remaining reagents and chemicals used in this study were purchased from Sigma-Aldrich.

### 2.2. Animals and Animal Experiments

Kunming mice (KM) were obtained from the Animal Center Laboratory of the Xi’an Jiaotong University, China. Male mice (8 weeks old) were housed in pathogen-free cages and supplied with food and water *ad libitum*, under a 12 h light/12 h dark photoperiod at 24 ± 2 °C and 70% humidity. All procedures involving animals were approved by the Institutional Animal Care and Use Committee of the Northwest A&F University (DK2022071).

Melatonin was dissolved in absolute ethanol, diluted with 0.9% NaCl to obtain an ethanol concentration of 5% and stored in the dark at 4 °C. Diquat solutions were prepared using 0.9% NaCl as a solvent. Male mice were randomly divided into four groups (*n* = 6 per group). The control group comprised mice only administered vehicle. In the diquat group, mice were intraperitoneally injected with diquat at a dose of 10 mg/kg b.w./day while melatonin group with 10 mg/kg b.w./day. Finally, in the diquat and melatonin groups, mice were simultaneously administered diquat and melatonin. All groups were treated for 28 consecutive days prior to euthanasia. In each group of males treated with diquat or/and melatonin, 12 males were mated with females and the number of offspring produced was recorded and statistically analyzed.

### 2.3. Assessment of Sperm Parameters

Cauda epididymal sperm was collected as previously reported [1]. In brief, the left cauda epididymides were cut into small pieces in prewarmed normal saline and gently squeezed to allow fluid to flow out. The sperm count (sperm/mL) was measured using a Neubauer hemocytometer. Giemsa staining was used to evaluate the sperm malformation rate by counting the number of morphological abnormalities in the head and flagellum and expressed as a percentage of the total sperm count.

### 2.4. Hematoxylin-Eosin (H&E) Staining and Quantitative Measurements of Spermatogenesis

The left testes and epididymides were fixed in a 4% (*w*/*v*) paraformaldehyde solution, embedded in paraffin blocks; three consecutive sections (5 μm) were then prepared and stained with hematoxylin and eosin (HE). Stained sections were observed using a Nikon optical microscope and images were captured for histological analyses. The germinal epithelium height of each seminiferous tubule was measured using the software Image Pro Plus 6.0. The germ cell including spermatogonia (Sg), preleptotene spermatocytes (PL), pachytene spermatocytes (P), and stages 1–7 (S1–7) spermatids were counted according to the method of a previous report [20,21]. All the crude counts of germ cells were corrected for section thickness and the differences in the nuclear or nucleolar diameter using Abercrombie’s formula, that is, P = A × [M /(L + M)] (P: average count of nuclear points per section; A: crude number of nuclei seen in the section; M: thickness (μm) of the section; L: average length (μm) of the nuclei) [22]. The corrected count of germ cells was used in the quantitative analysis.

### 2.5. Testosterone Assay

Serum testosterone levels were detected using a Testosterone Enzyme-Linked ImmunoSorbent Assay Kit (Cat#: PT872; Beyotime) and procedures were performed according to the manufacturer’s instructions. Briefly, the serum was incubated with the enzyme conjugate solution for 2 h at room temperature away from light in the ELISA plate. After washing the plate three times, the tetramethylbenzidine chromogen (TMB) was added and incubated for 20 min at room temperature in the dark. Then the stop solution was added and the absorbance was measured at 450 nm using a microplate reader (BioTek, Winooski, VT, USA).

### 2.6. Cell Culture and Treatment

The spermatogonia GC1-spg (Cat#: CL0600; Procell, Wuhan, China) were cultured in Dulbecco’s modified Eagle’s medium (DMEM) supplemented with 10% fetal bovine serum (FBS). The spermatogonia was treated with different concentrations of diquat (20, 40, 80 and 100 μM) for 24 h to detect cell cytotoxicity. In all experiments, the concentrations of dimethyl sulfoxide (DMSO) were diluted to below 1% (*v*/*v*).

### 2.7. Flow Cytometry Assay for Cell Apoptosis

Cells were evaluated using an Annexin V-FITC Apoptosis Detection Kit (Cat#: C1062M; Beyotime) and procedures were performed according to the manufacturer’s instructions. Briefly, treated cells were centrifugated and collected, then suspended in 500 μL of binding buffer. Subsequently, Annexin V-FITC and Propidium Iodide (PI) were added and incubated for 10 min at room temperature. The stained cells were detected using flow cytometry (Beckman Co., Miami, FL, USA) and analyzed using FlowJo Software (FlowJo, LLC., Ashland, OR, USA). 

### 2.8. Western Blotting

Right testicular tissue was prepared for analysis using RIPA Lysis Buffer (Cat#*:* P0013B; Beyotime) with 1 mM phenylmethanesulfonyl fluoride (Cat#: ST506; Beyotime). The total amount of extracted protein was quantified using the BCA protein assay kit (Cat#: PC0020; Solarbio, Beijing, China). For western blotting, equal protein samples (30 µg) were loaded and separated using 8–15% SDS-PAGE gel, and then transferred onto a nitrocellulose membrane (Cat#: YA1711; Solarbio). Subsequently, the membrane was blocked with 5% skim milk dissolved in TBST for 2 h at room temperature and incubated overnight at 4 °C with the primary antibodies listed in Appendix A. After washing with TBST and incubation for 1.5 h at room temperature with the secondary antibody (Cat#: SA00001; Proteintech, Rosemont, IL, USA), the probes were detected using electrochemiluminescence (ECL) reagents (Cat#: WBULS0100; Millipore, Boston, MA, USA), exposed using ChemiDoc XRS (Bio-Rad, Hercules, CA, USA). The blot intensity was quantified using Image Pro Plus 6.0.

### 2.9. Immunofluorescence Staining (IF)

The paraffin sections were permeabilized treated with 0.25% (*v*/*v*) Triton X-100 for 20 min at 37 °C, blocked with 5% BSA (*w*/*v*) for 30 min, and incubated overnight with ZO-1 (1:100) primary antibodies at 4 °C. The following day, the samples were washed with PBS and incubated with secondary antibodies at room temperature for 2 h in the dark, and then underwent nuclear staining using 4′,6-diamidino-2-phenylindole (DAPI). Images were acquired with a Nikon fluorescence microscope, and densitometric analysis was performed using Image Pro Plus 6.0.

### 2.10. Evaluation of ROS Levels 

The paraffin sections were incubated with 10 μM carboxy-2′,7′-dichloro-dihydro-fluorescein diacetate (DCFH-DA) (Cat#: S0033S; Beyotime) probe for 15 min at 37 °C. Subsequently, each section was washed with PBS three times for 10 min and stained with DAPI for 10 min. Images were captured using a Nikon fluorescence microscope (Nikon, Tokyo, Japan) and analyzed using Image Pro Plus 6.0.

Testicular oxidative stress levels were determined by measuring malondialdehyde (MDA) and using the total Superoxide Dismutase (SOD) (WST-1 method), and glutathione peroxidase (GSH-PX) (colorimetric method) assay kits (Nanjing Jiancheng Bioengineering Institute). All detection procedures were performed according to the manufacturer’s instructions. Briefly, testicular tissue was cut and homogenized using a Dounce homogenizer, and total protein was detected using a BCA protein assay kit (Cat#: PC0020; Solarbio). Testicular oxidative stress parameters MDA, SOD, and GSH-PX were evaluated using the corresponding kit, and the absorbance was measured using a microplate reader (BioTek).

### 2.11. TUNEL Assays

A Colorimetric TUNEL Apoptosis Assay Kit (Cat#: C1098; Beyotime) was used for terminal deoxynucleotidyl transferase dUTP nick-end labeling (TUNEL) assays. According to the manufacturer’s instructions, the paraffin sections were rehydrated, washed with distilled water, and incubated with proteinase K for 20 min at 37 °C. Sections were submerged in a 3% hydrogen peroxide solution for 10 min to block endogenous peroxidase activity. After washing thrice with PBS, the sections were balanced for 10 min and treated with the TUNEL reaction mixture at 37 °C for 1 h in a wet box. 

Subsequently, the sections were incubated for 30 min at 37 °C for staining with streptavidin-HRP and visualized with a freshly prepared 3,3′-diaminobenzidine (DAB) solution. Finally, nuclei counter-staining was conducted with hematoxylin e resulting nuclei were blue, TUNEL-positive cells were brown, and the ratio of TUNEL-positive cells to the total number of cells was analyzed to assess cell apoptosis.

### 2.12. Statistical Analysis

Data were presented as mean ± standard error of mean (SEM). Statistical significance was assessed using two-tailed Student’s *t*-test or one-way analysis of variance (ANOVA), followed by Duncan’s multiple range test with the statistical software SPSS V22.0 (IBM, Armonk, NY, USA). Results were considered statistically significant at *p* values < 0.05. At least three independent experiments were performed and quantified.

## 3. Results

### 3.1. Melatonin Ameliorates Spermatogenic Failure Induced by Diquat in Mice

To investigate testicular injuries and the role of melatonin in response to diquat treatment in mice, testicular and semen parameters were measured. During the 28 day-long treatment, we found that the weight of testes (Figure 1A,B) and epididymides (Figure 1E) were significantly lower than that of the control group. Similarly, both the testes/body (Figure 1C) and epididymides/body weight ratios (Figure 1F) were significantly decreased. According to the method of a previous report [21], histopathological analysis revealed that the diquat treatment shortened the germinal epithelium height (Figure 1D, Table 1) and reduced the number of spermatogonia (Sg), preleptotene spermatocytes (PL), pachytene spermatocytes (P), and stages 1–7 (S1–7) spermatids (Table 1) as compared with the control testis. Notably, diquat-treated cauda epididymides, i.e., the primary storage site for mature sperm, had a reduced density of mature sperm compared with that of the control group (Figure 1D). Meanwhile, the serum testosterone concentration reduced significantly induced by diquat (Figure 1H). As expected, exposure to diquat resulted in a highly significant decrease in sperm count (Figure 1G) and abnormal sperm morphology (Figure I) compared with control mice, suggesting that the spermatogenic function of the testes was compromised. Daily melatonin administration protected the mice from testicular damage caused by diquat. As shown in Figure 1, melatonin alleviated testis weight loss (Figure 1A,B) in mice and testicular injury, including inhibited spermatogenesis (Figure 1D, Table 1), increased sperm malformations (Figure 1I), and declined testosterone level (Figure 1H) when applied with diquat, whereas melatonin treatment alone had no effect on spermatogenesis.

Furthermore, the fertility of diquat-and/or melatonin-treated male mice was evaluated by mating them with untreated female mice. Melatonin increased the number of diquat-induced offspring, indicating that melatonin plays an important role in restoring the fertility of male mice exposed to diquat (Figure 1J).

### 3.2. Melatonin Protects Testis against Oxidative Stress Induced by Diquat

To evaluate diquat-induced oxidative stress in the testes, ROS levels and antioxidative indicators of testicular tissue were measured. As shown in Figure 2A, the fluorescence of the DCFH probe (a ROS indicator) significantly increased in diquat-treated testes (Figure 2B), as did the malonaldehyde (MDA) levels (Figure 2C). Treatment with melatonin markedly attenuated ROS and MDA levels (Figure 2A–C). 

Furthermore, the activities of the antioxidant enzymes SOD and GSH-Px in the testis were significantly reduced in the diquat-treated group, which were ameliorated by melatonin treatment (Figure 2D,E). Melatonin treatment also significantly restored the decreased expression of GPX1 and SOD1 caused by diquat (Figure 2F). Hence, melatonin protected testicles from the diquat-induced oxygen stress response by increasing the abundance of the antioxidant enzymes, GPX1 and SOD1. 

### 3.3. Melatonin Attenuates Diquat-Induced Apoptosis in the Mouse Testes

It is well known that elevated ROS levels induce apoptosis. Figure 3A,B shows an increase in the number of TUNEL-positive cells in the seminiferous tubules of the diquat-treated group, which was alleviated by melatonin treatment (Figure 3A,B). Furthermore, we investigated the expression of apoptosis-related markers, P53, Cleaved-Caspase3, Bax, and Bcl2 by Western blotting and found that melatonin reversed the upregulation of the pro-apoptotic protein BAX and downregulation of the anti-apoptotic BCL2, induced by diquat (Figure 3C). Furthermore, the expression of Cleaved-Caspase 3, an effector of the mitochondria-mediated apoptosis pathway, as well as the apoptotic activating protein P53, were inhibited by melatonin treatment (Figure 3C). These findings suggest that melatonin inhibits diquat-induced testicular apoptosis.

As shown in Figure 3A, more TUNEL-positive spermatogonial stem cells were observed in the diquat treated group than in the other two groups, indicating that spermatogonial damage was most pronounced in the testicular tissue. As a result, we then detected the apoptosis rate of the GC1-spg cell, a type of spermatogonia, exposed to diquat via the flow cytometry assay. The results showed that diquat significantly induced spermatogonia cell apoptosis (Appendix A). Furthermore, the expression of Cleaved-Caspase3 and Bax/Bcl2 of the spermatogonia cells was markedly increased in those treated with diquat compared with the control group (Appendix A). These findings confirm diquat-induced spermatogonial cytotoxicity through the apoptotic pathway. 

### 3.4. Melatonin Restores the Diquat-Disrupted Integrity of the Blood-Testis Barrier 

Sertoli cells orchestrate spermatogenesis by maintaining the spermatogonial stem cell niche and spermatogonial populations [13,23,24]. Therefore, we hypothesized that melatonin could prevent diquat-induced testicular damage by maintaining the integrity of Sertoli cell junctions, which forms the BTB. As expected, immunofluorescence of ZO-1 exhibited a severely fractured and disordered staining pattern in the seminiferous epithelium, indicating that diquat damaged the Sertoli cell development and function. Meanwhile, melatonin treatment sustained the normal expression of ZO-1 (Figure 4A,B). We also measured the levels of several major adhesion junction, gap junction, and tight junction proteins and found that expression levels of β-catenin, a major component of adhesion junctions, as well as occludin, another major component of tight junctions, such as ZO-1, were reduced by diquat treatment and were restored by melatonin treatment (Figure 4C). Expression of the main gap junction protein Connexin43 did not significantly change following diquat and/or melatonin treatment (Figure 4C). These results indicate that melatonin restored the diquat-disrupted integrity of BTB, thereby protecting mouse testes against diquat-induced damage. 

## 4. Discussion

Diquat is a non-selective bipyridyl herbicide widely used in agriculture in various regions of the world [25]. A survey study showed that in addition to intentional or unintentional ingestion, 29% of acute illnesses associated with diquat are work-related, and caused by inadequate use of personal protective equipment and herbicide spraying/splashing in agricultural applications [1]. Moreover, exposure to diquat corrodes the skin and gastrointestinal tract [26] and damages the kidneys, liver, heart, and central nervous system [25]. Meanwhile, paraquat, another dipyridyl herbicide, can induce deleterious changes in mammalian testis [27,28,29,30]. However, diquat is reportedly less potent in male reproductive impairment. In this work, 28 days of exposure to diquat resulted in testicular damage in mice, including damaged seminiferous epithelium with less germ cells, decreased sperm count, and increased sperm deformity. Moreover, the serum testosterone level declined significantly, which was induced by diquat.

Melatonin, an indoleamine synthesized primarily by the pineal gland, reproductive organs, testes, and ovaries, modulates multiple physiological functions in mammals [9,10,11]. For example, melatonin preserves sperm quality by regulating testosterone production in Leydig cells [12,31], modulates proliferation and energy metabolism in Sertoli cells [13], and protects sperm from free radical damage during their passage through the reproductive tract [32]. In addition, as a powerful free-radical scavenger, melatonin relieves testicular damage through their antioxidant and anti-apoptotic properties [14,33]. Our findings are consistent with these reports, suggesting that melatonin rescues testicular dysfunction and ameliorates spermatogenic failure induced by diquat.

The current study also revealed that diquat induced the upregulation of ROS and the downregulation of antioxidants. Similarly, previous animal models have demonstrated that diquat induces oxidative stress [34,35,36] due to its powerful capacity to produce superoxide anion free radicals [37]. Meanwhile, melatonin is essential for maintaining redox balance in the testes [38,39]. An imbalance between ROS and the antioxidant response system can lead to serious oxidative stress. Thus, our study confirms that melatonin effectively inhibited ROS generation. Furthermore, the diquat-induced reduction in GPX1 and SOD1 protein abundance, was reversed following treatment of mice with melatonin, a result that is in accordance with previous studies [40,41,42,43]. 

Studies have demonstrated that diquat is involved in mitochondrial dysfunction and apoptosis [44,45,46], although Nisar [47] found that diquat damages neural tissue by programmed necrosis rather than by apoptosis. Thus, we hypothesized that diquat-induced ROS production can cause apoptosis in testicular cells of mice. This was confirmed by TUNEL staining, which showed an increased number of apoptotic cells. Particularly, spermatogonia was most damaged by diquat in an apoptotic manner in vivo and in vitro. In contrast, melatonin treatment reduced TUNEL-positive cells, which agrees with the report that melatonin can protect the testes from external harm via a synergistic interaction of its antioxidant, anti-inflammatory, and anti-apoptotic properties [17]. This was also confirmed by changes in Cleaved-Caspase3 and Bax/Bcl2 levels induced by melatonin treatment. Furthermore, we found that P53, a key effector gene that induces apoptosis [48], was upregulated following diquat treatment, causing apoptosis, whereas melatonin treatment attenuated this effect. These findings are consistent with those of similar studies [45,46].

The BTB, comprising the junctions between Sertoli cells, can provide a relatively enclosed microenvironment favoring germ cell survival and spermatogenesis. Hence, maintained BTB integrity is crucial for male reproduction [13,49]. Previous studies have reported that heat stress damages the integrity of Sertoli cells and causes spermatogenesis failure [15,50]. Meanwhile, another herbicide, flurochloridone, influences testicular function via inducing mitochondrial damage and apoptosis of testicular Sertoli cells [51]. We, therefore, evaluated the localization of ZO-1 by immunofluorescence. Moreover, considering that tight junctions and adhesion junctions are required for the junctions between Sertoli cells, and between Sertoli cells and germ cells, respectively [52,53], we also assessed the levels of several major adhesion junction, gap junction, and tight junction proteins. Interestingly, the abundance of tight junction proteins, occludin and ZO-1, was reduced. Similarly, β-catenin, a major component of adhesion junctions, was inhibited by diquat. Hence, given that the WNT/β-catenin signaling pathway contributes to the stimulation of other processes, such as cell proliferation and differentiation [54,55], further investigation into the diquat-induced effects on β-catenin is warranted. The observed diquat-induced destruction of Sertoli cell tight junctions is consistent with the findings of a study reporting that diquat inhibits the expression of occludin and ZO-1, resulting in intestinal barrier function damage in piglets [35]. Melatonin treatment rescued the tight junction expression and improved their order of localization. Similarly, melatonin has been show to impact spermatogenesis by modulating Sertoli cell metabolism [13], and specifically maintains the integrity of the BTB preventing injury [15]. 

Herein, we preliminarily explored the protective effect of melatonin on the reproductive toxicity of diquat, as well as the associated molecular mechanisms; however, further investigation into the associated mechanisms within the testicular cells are warranted. Moreover, although it is clear that melatonin exhibits potent receptor-dependent and -independent actions, including antioxidant, anticancer, antitumor, anti-inflammatory, anti-aging, anti-diabetic, antiviral, and neuroprotective activities [56,57], and we posit that its direct free-radical scavenging actions are receptor-independent; however, it still remains to be determined whether diquat affects testis injury in a melatonin receptor-dependent manner. Indeed, previous studies have shown that melatonin treatment induces the expression of SIRT1, thereby reducing ROS levels and thus ameliorating oocyte aging and palmitic acid-induced testis lipotoxicity [58,59]. Furthermore, several melatonin-related effects can be abolished via SIRT1 inhibition, indicating potential mediation by SIRT1 in non-tumor cells [60]. Hence, it is necessary to also verify whether melatonin protects the testes from diquat-induced oxidative stress by regulating SIRT1. Meanwhile, other studies suggest that diquat induces lipid peroxidation [61], in which hepatocytes and intestinal mucosa ferroptosis were caused following the treatment of piglets for seven days, via regulating the expression of ferroptosis mediators (transferrin receptor protein 1, heat shock protein beta 1, solute carrier family 7 member 11, and glutathione peroxidase 4) [62,63,64]. It would, therefore, be of interest to determine whether melatonin reverses testis injury following diquat exposure in a ferroptosis-associated manner.

## 5. Conclusions

Taken together, the findings of this study demonstrate that melatonin protects testes from diquat-induced oxidative stress and apoptosis, ensuring the integrity of the BTB in mice. Hence, this study provides a theoretical basis for the potential application of melatonin as a preventive or therapeutic drug for the treatment of male sub- or infertility in populations with high exposure risk to diquat.

## Figures and Tables

**Figure 1 toxics-11-00160-f001:**
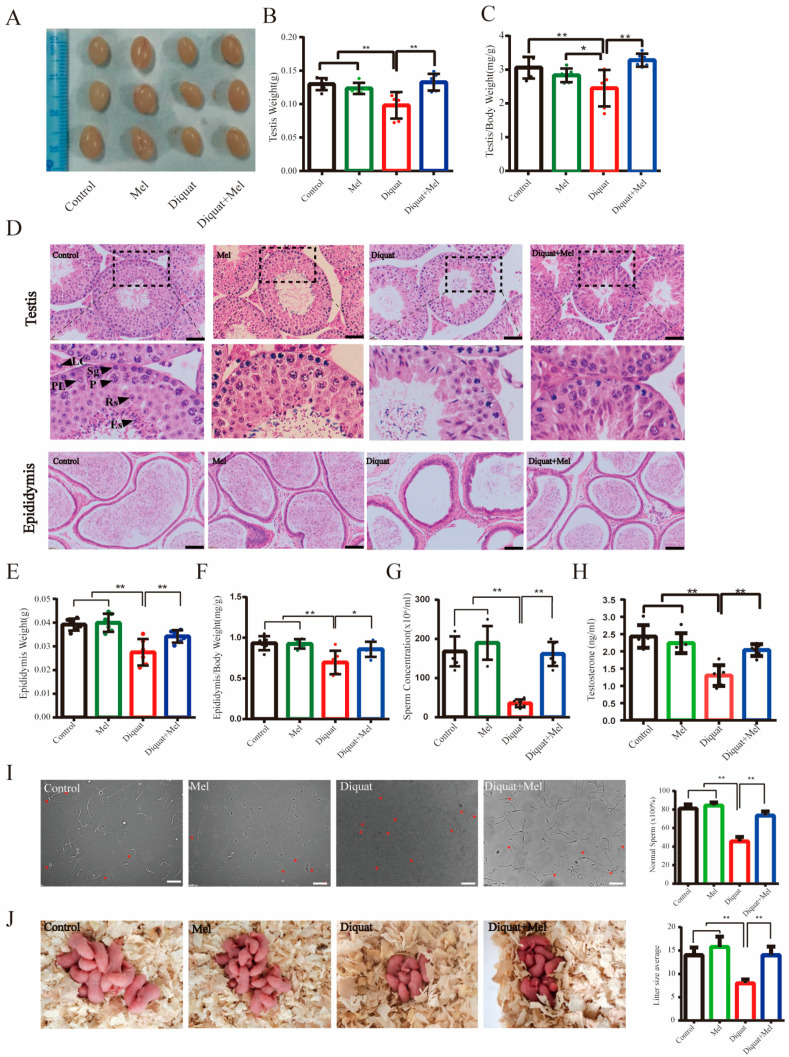
Melatonin ameliorates spermatogenic failure induced by diquat in mice. Male mice were intraperitoneally injected with ethanol (control), melatonin (10 mg/kg/d), diquat (10 mg/kg/d), or diquat combined with melatonin for 28 consecutive days: (**A**) image of testes; (**B**) weights of testes and (**E**) epididymides; (**C**) organ coefficients of testes; and (**F**) epididymides in respective treatment groups; (**D**) morphology of testes and epididymides in the different groups. The squares represent the enlarged position in the following figure. Sp: spermatogonium; PL: preleptotene spermatocyte; P: pachytene spermatocyte; Rs: round spermatids; Es: elongated spermatids; LC: Leydig cells. Scale bars = 50 μm; (**G**) sperm concentration in the cauda epididymides of mice under different treatments; (**H**) serum testosterone concentration in different treated groups; (**I**) sperm morphology and the percentage of normal sperm per epididymis. Red arrowheads denote abnormal sperm. Scale bars = 50 μm; and (**J**) the graph and number of offspring in different treatments males. Data are presented as the mean ± SEM of at least three independent experiments. * *p* < 0.05, ** *p* < 0.01.

**Figure 2 toxics-11-00160-f002:**
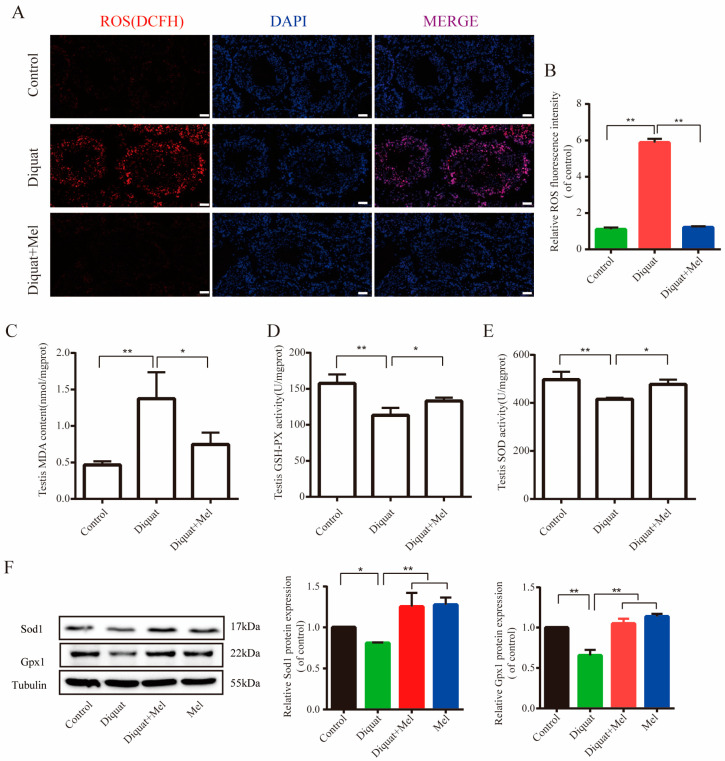
Melatonin protects testis against oxidative stress caused by diquat: (**A**) representative images of ROS levels in the control and diquat treatment with or without melatonin; (**B**) relative fluorescence density of DCFH analyzed using Image J. Scale bars = 10 μm; (**C**–**E**) analysis of the parameters for oxidative stress in the differently treated testes; and (**F**) representative Western blotting of the related antioxidant enzymes SOD1 and GPX1 of the control, melatonin (10 mg/kg/d), diquat (10 mg/kg/d), and diquat combined with melatonin-treated groups. Data are presented as the mean ± SEM of at least three independent experiments. * *p* < 0.05, ** *p* < 0.01. MDA, malondialdehyde; GSH-PX, glutathione peroxidase; SOD, superoxide dismutase.

**Figure 3 toxics-11-00160-f003:**
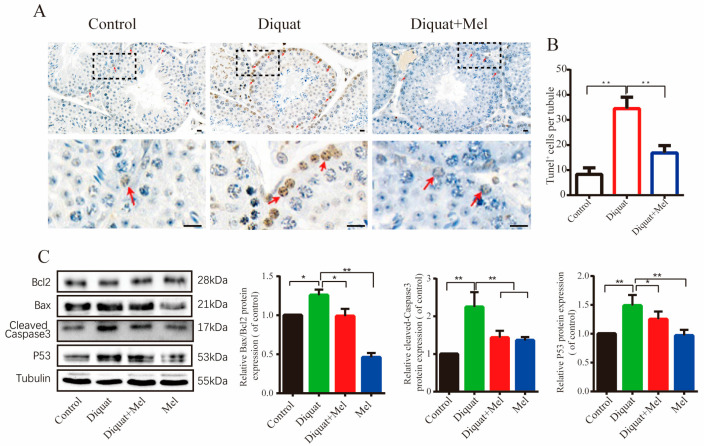
Melatonin attenuates diquat-induced apoptosis in mouse testes: (**A**) representative TUNEL staining photographs of testes in the control and diquat treatment groups with or without melatonin. TUNEL-positive cells denoted by arrowheads are presented in brown while the nucleus is stained blue. The squares represent the enlarged position in the following figure. Scale bars = 5 μm; (**B**) quantitative analysis of the number of apoptotic cells; and (**C**) representative Western blotting and quantitative analysis of apoptosis-related markers, Bax/Bcl2, Cleaved-Caspase3, and P53 in the control, melatonin (10 mg/kg/d), diquat (10 mg/kg/d) or diquat combined with melatonin treatment groups. Data are presented as the mean ± SEM of at least three independent experiments. * *p* < 0.05, ** *p* < 0.01.

**Figure 4 toxics-11-00160-f004:**
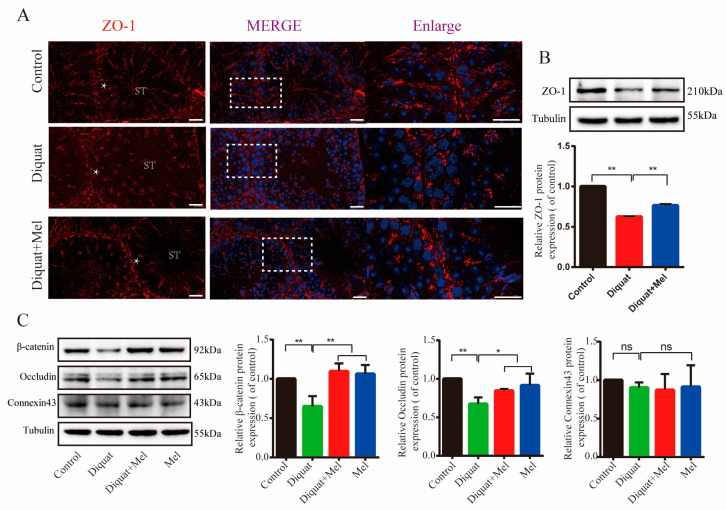
Melatonin restores the diquat-disrupted integrity of Sertoli cells: (**A**) representative immunofluorescence images showing the expression and localization of ZO-1 protein (The squares represent the enlarged position in the following figure); (**B**) relative expression of ZO1 using Western blotting in the control and diquat treatment with or without melatonin groups. Scale bars = 10 μm; and (**C**) representative Western blotting and quantitative analysis of β-catenin, Occludin and Connexin43 in the different treatment groups. ST: seminiferous tubule. Asterisks indicate Sertoli cells. Data are presented as the mean ± SEM of at least three independent experiments. ns: not significant (*p* ≥ 0.05), * *p* < 0.05, ** *p* < 0.01.

**Table 1 toxics-11-00160-t001:** Melatonin ameliorates thinness of seminiferous epithelium and loss of germ cells induced by diquat in mice.

Treatment Groups	Germinal Epithelium Height (μm)	Spermatogonia	Preleptotene Spermatocyte	Pachytene Spermatocyte	Stages 1–7 Spermatids
Control	76.92 ± 2.91 ^b^	19.30 ± 0.71 ^b^	8.33 ± 0.34 ^b^	9.86 ± 0.20 ^b^	66.80 ± 4.77 ^b^
Mel	69.23 ± 5.81 ^b^	18.19 ± 0.52 ^b^	9.58 ± 1.04 ^b^	9.58 ± 0.90 ^b^	64.44 ± 5.21 ^b^
Diquat	39.74 ± 6.05 ^a^	9.02 ± 1.19 ^a^	5.69 ± 0.52 ^a^	7.22 ± 0.52 ^a^	45.28 ± 2.39 ^a^
Diquat + Mel	67.95 ± 4.44 ^b^	17.80 ± 0.55 ^b^	7.78 ± 0.71 ^b^	9.30 ± 0.85 ^b^	66.53 ± 3.41 ^b^

Note: Data are represented as the mean percentage ± SEM. Values within a row with different letters. (^a^, ^b^) indicate significant differences (*p* < 0.05).

## Data Availability

Data is contained within the article and Appendix A.

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
