# Peer review of "Melatonin Ameliorates Diquat-Induced Testicular Toxicity via Reducing Oxidative Stress, Inhibiting Apoptosis, and Maintaining the Integrity of Blood-Testis Barrier in Mice"

_toxics, 2023, doi:10.3390/toxics11020160_

Round 1

Reviewer 1 Report

The Yang et al., 2022, manuscript ID toxics-2176540 investigated the effect of melatonin in ameliorating diquat-induced testicular toxicity and oxidative stress via inhibiting testicular apoptosis, and maintaining the integrity of blood-testis barrier in the mice. There are few queries and few suggestions which makes this manuscript more representable to be publish.

Major Comments

1.      The authors claiming that they have assessed the molecular mechanism of Diquat-induced testicular damage, but it is not clear the cellular mechanism of melatonin. The authors need to show some experiments to prove that which testicular cells got damaged by Diquat. The molecular mechanism is not getting clear as the authors didn’t show the testosterone levels in the serum or steroidogenic enzymes in the testis of the mice which is a crucial indicator of testicular steroidogenesis. Did they find any histological changes in the Leydig cells?

2.      Do the authors find any changes in the number of spermatogenic cells. They can count the number of spermatogenic cells (Type A spermatogonia, preleptotene (PL) spermatocyte, Pachytene (P) spermatocyte, Stages 1–7 spermatids), Germinal epithelium height (μm) etch. They can refer the article PMID: 30471430.

Minor Comments

Serious spelling mistake at several places throughout the MS. English correction and sentence reframing have to be performed.

1       Proper labelling of testicular histological cells must be done?  

3.      In the Line 192…. The graph and number of offspring of males in 192 different treatments. All offspring showing in the result section are male. How many breading pair resulted in this litter size?

4.      Did the authors found any changes in the testicular and serum glucose and insulin level?

5.      Line 300…. R. Nisar This is not the way to mentioned in the text. It need to be corrected.

6.      Why the authors have not check the activities of anti-oxidative enzymes in the treated groups?

Author Response

January 23, 2023

Dear reviewer:

        We appreciate the reviewer’s invaluable comments and suggestions for the manuscript (Toxics-2176540). Those comments are all valuable and very helpful for revising and improving our manuscript. We have studied comments carefully and have made corrections which we hope meet with approval. All the corrections in the manuscript and the responds to the reviewer’s comments are as follows.

Point-by-point response to the reviewers’ comments

Major Comments

1. The authors claiming that they have assessed the molecular mechanism of Diquat-induced testicular damage, but it is not clear the cellular mechanism of melatonin. The authors need to show some experiments to prove that which testicular cells got damaged by Diquat. The molecular mechanism is not getting clear as the authors didn’t show the testosterone levels in the serum or steroidogenic enzymes in the testis of the mice which is a crucial indicator of testicular steroidogenesis. Did they find any histological changes in the Leydig cells?

Response 1: We appreciate the reviewer’s invaluable suggestion. As shown in Figure 3 A, more TUNEL-positive spermatogonial stem cells were observed in the Diquat treated group, indicating that spermatogonial damage was most pronounced in the testicular tissue. Subsequently, the toxic effects of Diquat on spermatogonia were verified using GC-1 spg cell line, including apoptosis rate via the flow cytometry assay and the expression of Bax, BCL2, cleaved-Caspase3 pretein as follows(Figure S1). The level of serum testosterone was also measured(Figure 1H). An attempt was made to demonstrate the mechanism by which Diquat damages the testis. Furthermore, we did not find significant histological changes in Leydig cell(Figure 1D).

2. Do the authors find any changes in the number of spermatogenic cells. They can count the number of spermatogenic cells (Type A spermatogonia, preleptotene (PL) spermatocyte, Pachytene (P) spermatocyte, Stages 1–7 spermatids), Germinal epithelium height (μm) etch. They can refer the article PMID: 30471430.

Response 2: Thank you for reviewer’suggestion. We have labed the spermatogenic cells in Figure 1D , counted the number of Spermatogonia(Sg), preleptotene (PL) spermatocyte, Pachytene (P) spermatocyte, Stages 1–7 spermatids and measureed the germinal epithelium height in Table1.

Minor Comments

Serious spelling mistake at several places throughout the MS. English correction and sentence reframing have to be performed.

Response: Thank you for raising this point. We have had the manuscript edited by a professional English editor at Editage to ensure there are no remaining spelling, grammatical, or linguistic errors. A certificate confirming that English editing was performed was obtained and will be forwarded to the Editorial Office.

1. Proper labelling of testicular histological cells must be done?

Response 1: As reviewer’suggestion, testicular histological cells had been proper labed in Figure 1D and Figure 4A.

2. In the Line 192…. The graph and number of offspring of males in 192 different treatments. All offspring showing in the result section are male. How many breeding pair resulted in this litter size?

Response 2We apologize for our negligence and unclear description. We have revised the text for clarity to indicate that in each group of males treated with diquat or/and melatonin, 12 males were mated with females. The number of offspring including male and female pups were recorded and statistically analyzed. We have updated the image presented in the original Figure 1 (J,K) to “(J) The graph and number of offspring from males under different treatments.”

3. Did the authors found any changes in the testicular and serum glucose and insulin level?

Response 3Thank you for this query. Indeed, diquat is not only a non-selective bipyridyl herbicide widely used in agriculture but also an oxidative stress inducer. However, our study focused primarily on protective effect of melatonin on the testes against diquat-induced damage via reducing oxidative stress, inhibiting apoptosis, and maintaining the association between Sertoli cells. As such, we did not included metabolic studies as they were deemed outside the scope of the current study. Nevertheless, this is certainly an interesting aspect to investigate in future studies.

4. Line 300…. R. Nisar This is not the way to mentioned in the text. It need to be corrected.

Response 4Thank you for bringing this to our attention. We have revised the reference format to “Studies have demonstrated that diquat is involved in mitochondrial dysfunction and apoptosis [1-3], although Nisar [4] found that diquat damages neural tissue by programmed necrosis rather than by apoptosis.”

5. Why the authors have not check the activities of anti-oxidative enzymes in the treated groups?

Response 5In Figure 2C–E, we have evaluated the activity of testicular anti-oxidant enzymes, namely, superoxide dismutase (SOD), glutathione peroxidase (GSH-PX), and malondialdehyde (MDA). The activities of SOD and GSH-Px were significantly reduced within the testis of the diquat-treated group, however, this effect was ameliorated by melatonin treatment. Moreover, melatonin markedly attenuated the increase in malonaldehyde (MDA) induced by diquat.

References

  1. Jin, Y.; Zhai, Z.; Jia, H.; Lai, J.; Si, X.; Wu, Z. Kaempferol attenuates diquat-induced oxidative damage and apoptosis in intestinal porcine epithelial cells. Food Funct 2021, 12, 6889-6899, doi:10.1039/d1fo00402f.
  2. Park, A.; Koh, H.C. NF-kappaB/mTOR-mediated autophagy can regulate diquat-induced apoptosis. Arch Toxicol 2019, 93, 1239-1253, doi:10.1007/s00204-019-02424-7.
  3. Choi, S.E.; Park, Y.S.; Koh, H.C. NF-kappaB/p53-activated inflammatory response involves in diquat-induced mitochondrial dysfunction and apoptosis. Environ Toxicol 2018, 33, 1005-1018, doi:10.1002/tox.22552.
  4. Nisar, R.; Hanson, P.S.; He, L.; Taylor, R.W.; Blain, P.G.; Morris, C.M. Diquat causes caspase-independent cell death in SH-SY5Y cells by production of ROS independently of mitochondria. Arch Toxicol 2015, 89, 1811-1825, doi:10.1007/s00204-015-1453-5.

Reviewer 2 Report

The study ‘Melatonin ameliorates diquat-induced testicular toxicity via reducing oxidative stress, inhibiting apoptosis, and maintaining 3 the integrity of blood-testis barrier in mice’ is interesting but still authors need to address following points.

Line 45 please correct zebra fish spelling

Figure 1G, It is evident from the data that melatonin increased the concentrations of sperms, then how come does the authors claim that melatonin does not affect the spermatogenesis process alone? As the authors have summarized the figure-1 as ‘As shown in Figure 1, melatonin alleviated testis weight loss in mice and effectively prevented spermatogenic failure when applied with diquat, whereas melatonin treatment alone had no effect on spermatogenesis’. Please clarify

Figure 2F-H represent western blot analysis so these all three can be merged as Figure 2F.

Line 245 is missing a reference of the statement.

Again, in Figure-4, Western blot images and their respective graphs represent the same data… No need to represent them separately Figure 4B and 4C represent 4B…. Both images can be names as one…. The same has happened with other images in the group. Please rectify

Anti-oxidant and anti-inflammatory activities of Melatonin in testicular toxicity has been done in a number of other studies. So, what makes this study unique. Please state

Please state the limitations of the current study

Author Response

January 23, 2023

Dear reviewer:

We thank you for your thoughtful suggestions and comments on the manuscript (Toxics-2176540). Those comments are all valuable and very helpful for revising and improving our manuscript, as well as the important guiding significance to our researches.The manuscript has been rechecked and the necessary changes have been made in accordance with the reviewer’ s suggestions. We hope that the responses have addressed all the reviewers’ concerns satisfactorily. All the corrections in the manuscript and the responds to the reviewer’s comments are as follows.

Point-by-point response to the reviewers’ comments

Comments from the Reviewers #2:

1. Line 45 please correct zebra fish spelling

Response 1Thank you for this comment. However, in line 45, “Research involving adult zebra finch males showed that diquat treatment led to a decline in sperm velocity and a shorter sperm midpiece.”, the term “zebra finch” actually refers to “Taeniopygia guttata”, which is a bird species rather than the zebra fish. Nevertheless, the text has been revised to “Research involving adult zebra finch (Taeniopygia guttata) males showed that diquat treatment led to a decline in sperm velocity and a shorter sperm midpiece.” to avoid confusion.

2. Figure 1G, It is evident from the data that melatonin increased the concentrations of sperms, then how come does the authors claim that melatonin does not affect the spermatogenesis process alone? As the authors have summarized the figure-1 as ‘As shown in Figure 1, melatonin alleviated testis weight loss in mice and effectively prevented spermatogenic failure when applied with diquat, whereas melatonin treatment alone had no effect on spermatogenesis’. Please clarify

Response2: We appreciate the reviewer’s invaluable suggestion. As shown in Figure 1G, although there was a trend toward “melatonin increasing the concentration of sperm”, statistically, there was no significant difference in the melatonin treatment alone group compared to the control group. Moreover, mechanistically, we found that melatonin exerts efficient antioxidant effects in the presence of an oxidative stimuli, to balance oxidation and antioxidation processes. In the current study, melatonin was shown to protect the testes from diquat-induced damage by reducing oxidative stress, inhibiting apoptosis, and maintaining the association between Sertoli cells, however, melatonin did not exhibit significant antioxidant activity in the absence of diquat in the mouse testes.

3. Figure 2F-H represent western blot analysis so these all three can be merged as Figure 2F. Again, in Figure-4, Western blot images and their respective graphs represent the same data… No need to represent them separately Figure 4B and 4C represent 4B…. Both images can be names as one…. The same has happened with other images in the group. Please rectify.

Response3: Thank you for raising this point and for your suggestion. The redundant data in the original Figure 1H, I has been merged into the revised Figure 1H; Figure 1J,K has been merged into Figure 1J; the same western blot analysis presented in Figure 2F–H has been merged into the revised Figure 2F; the original Figure 3C–F has been merged into Figure 3C; Figure 4B,C has been merged to Figure 4B; Figure 4D–G have been merged into Figure 4C.

4.Line 245 is missing a reference of the statement.

Response 4: Appropriate references have been added to line 248, as follows:

“Sertoli cells orchestrate spermatogenesis by maintaining the spermatogonial stem cell niche and spermatogonial populations [1-3].”

5. Again, in Figure-4, Western blot images and their respective graphs represent the same data… No need to represent them separately Figure 4B and 4C represent 4B…. Both images can be names as one…. The same has happened with other images in the group. Please rectify

Response 5: Please refer to our response to Comment 3.

6. Anti-oxidant and anti-inflammatory activities of Melatonin in testicular toxicity has been done in a number of other studies. So, what makes this study unique. Please state

Response 6: We appreciate the reviewer’s invaluable suggestion. As discussed in our study, melatonin relieves testicular damage caused by hyperthermia [4], irradiation [5,6], and environmental toxins [7,8].” However, no studies have reported on the male reproductive toxicity of diquat in mammals, nor on the role of melatonin in countering diquat effects on male fertility. As such, our research provides new insights regarding the action of a natural antioxidant, melatonin, against diquat-induced damage to male fertility in mice, and a theoretical basis for future research to counter herbicide genotoxic effects in mammals.

7. Please state the limitations of the current study

Response 7: Thank you for this comment. Here is the limitations of our study we thought.“Herein, we preliminarily explored the protective effect of melatonin on reproductive toxicity of diquat, as well as the associated molecular mechanisms; however, further investigation into the associated mechanisms within the testicular cells are warranted. Moreover, although it is clear that melatonin exhibits potent receptor-dependent and-independent actions, including antioxidant, anticancer, antitumor, anti-inflammatory, anti-aging, anti-diabetic, antiviral, and neuroprotective activities [9,10], and we posit that its direct free radical scavenging actions are receptor independent, however, it remains to be determined whether diquat affects testis injury in a melatonin receptor-dependent manner. Indeed, previous studies have shown that melatonin treatment induces the expression of SIRT1, thereby reducing ROS levels, thus, ameliorating oocyte aging and palmitic acid-induced testis lipotoxicity [11,12]. Furthermore, several melatonin-related effects are abolished via SIRT1 inhibition, indicating potential mediation by SIRT1 in non-tumor cells [13]. Hence, it is necessary to also verify whether melatonin protect the testes from diquat-induced oxidative stress by regulating SIRT1. Meanwhile, other studies suggest that diquat induces lipid peroxidation [14], thereby hepatocytes and intestinal mucosa ferroptosis following treatment of piglets for 7 days, via regulating the expression of ferroptosis mediators (transferrin receptor protein 1, heat shock protein beta 1, solute carrier family 7 member 11, and glutathione peroxidase 4) [15-17]. It would, therefore, be of interest to determine whether melatonin reverses testis injury following diquat exposure in a ferroptosis-associated manner.”

References

  1. Rocha, C.S.; Martins, A.D.; Rato, L.; Silva, B.M.; Oliveira, P.F.; Alves, M.G. Melatonin alters the glycolytic profile of Sertoli cells: implications for male fertility. Mol Hum Reprod 2014, 20, 1067-1076, doi:10.1093/molehr/gau080.
  2. Wu, S.; Yan, M.; Ge, R.; Cheng, C.Y. Crosstalk between Sertoli and Germ Cells in Male Fertility. Trends Mol Med 2020, 26, 215-231, doi:10.1016/j.molmed.2019.09.006.
  3. O'Donnell, L.; Smith, L.B.; Rebourcet, D. Sertoli cells as key drivers of testis function. Semin Cell Dev Biol 2022, 121, 2-9, doi:10.1016/j.semcdb.2021.06.016.
  4. Zhang, P.; Zheng, Y.; Lv, Y.; Li, F.; Su, L.; Qin, Y.; Zeng, W. Melatonin protects the mouse testis against heat-induced damage. Mol Hum Reprod 2020, 26, 65-79, doi:10.1093/molehr/gaaa002.
  5. Take, G.; Erdogan, D.; Helvacioglu, F.; Goktas, G.; Ozbey, G.; Uluoglu, C.; Yucel, B.; Guney, Y.; Hicsonmez, A.; Ozkan, S. Effect of melatonin and time of administration on irradiation-induced damage to rat testes. Braz J Med Biol Res 2009, 42, 621-628, doi:10.1590/s0100-879x2009000700006.
  6. Amer, M.E.; Othman, A.I.; Abozaid, H.M.; El-Missiry, M.A. Utility of melatonin in mitigating ionizing radiation-induced testis injury through synergistic interdependence of its biological properties. Biol Res 2022, 55, 33, doi:10.1186/s40659-022-00401-6.
  7. Kim, S.H.; Lee, I.C.; Baek, H.S.; Shin, I.S.; Moon, C.; Kim, S.H.; Yun, W.K.; Nam, K.H.; Kim, H.C.; Kim, J.C. Melatonin prevents gentamicin-induced testicular toxicity and oxidative stress in rats. Andrologia 2014, 46, 1032-1040, doi:10.1111/and.12191.
  8. Li, B.; He, X.; Zhuang, M.; Niu, B.; Wu, C.; Mu, H.; Tang, F.; Cui, Y.; Liu, W.; Zhao, B.; et al. Melatonin Ameliorates Busulfan-Induced Spermatogonial Stem Cell Oxidative Apoptosis in Mouse Testes. Antioxid Redox Signal 2018, 28, 385-400, doi:10.1089/ars.2016.6792.
  9. Tamura, H.; Jozaki, M.; Tanabe, M.; Shirafuta, Y.; Mihara, Y.; Shinagawa, M.; Tamura, I.; Maekawa, R.; Sato, S.; Taketani, T.; et al. Importance of Melatonin in Assisted Reproductive Technology and Ovarian Aging. Int J Mol Sci 2020, 21, doi:10.3390/ijms21031135.
  10. Habtemariam, S.; Daglia, M.; Sureda, A.; Selamoglu, Z.; Gulhan, M.F.; Nabavi, S.M. Melatonin and Respiratory Diseases: A Review. Curr Top Med Chem 2017, 17, 467-488, doi:10.2174/1568026616666160824120338.
  11. Yang, Q.; Dai, S.; Luo, X.; Zhu, J.; Li, F.; Liu, J.; Yao, G.; Sun, Y. Melatonin attenuates postovulatory oocyte dysfunction by regulating SIRT1 expression. Reproduction 2018, 156, 81-92, doi:10.1530/REP-18-0211.
  12. Xu, D.; Liu, L.; Zhao, Y.; Yang, L.; Cheng, J.; Hua, R.; Zhang, Z.; Li, Q. Melatonin protects mouse testes from palmitic acid-induced lipotoxicity by attenuating oxidative stress and DNA damage in a SIRT1-dependent manner. J Pineal Res 2020, 69, e12690, doi:10.1111/jpi.12690.
  13. Hardeland, R. Aging, Melatonin, and the Pro- and Anti-Inflammatory Networks. Int J Mol Sci 2019, 20, doi:10.3390/ijms20051223.
  14. Zhang, L.; Wei, W.; Xu, J.; Min, F.; Wang, L.; Wang, X.; Cao, S.; Tan, D.X.; Qi, W.; Reiter, R.J. Inhibitory effect of melatonin on diquat-induced lipid peroxidation in vivo as assessed by the measurement of F2-isoprostanes. J Pineal Res 2006, 40, 326-331, doi:10.1111/j.1600-079X.2005.00311.x.
  15. Hua, H.; Xu, X.; Tian, W.; Li, P.; Zhu, H.; Wang, W.; Liu, Y.; Xiao, K. Glycine alleviated diquat-induced hepatic injury via inhibiting ferroptosis in weaned piglets. Anim Biosci 2022, 35, 938-947, doi:10.5713/ab.21.0298.
  16. He, P.; Hua, H.; Tian, W.; Zhu, H.; Liu, Y.; Xu, X. Holly (Ilex latifolia Thunb.) Polyphenols Extracts Alleviate Hepatic Damage by Regulating Ferroptosis Following Diquat Challenge in a Piglet Model. Front Nutr 2020, 7, 604328, doi:10.3389/fnut.2020.604328.
  17. Xu, X.; Wei, Y.; Hua, H.; Jing, X.; Zhu, H.; Xiao, K.; Zhao, J.; Liu, Y. Polyphenols Sourced from Ilex latifolia Thunb. Relieve Intestinal Injury via Modulating Ferroptosis in Weanling Piglets under Oxidative Stress. Antioxidants (Basel) 2022, 11, doi:10.3390/antiox11050966.

Round 2

Reviewer 1 Report

The authors have tried to convince me with the experiments suggested by me. But the manuscript still has flaws. The authors have to rewrite the abstract as it has some mistakes which needs to be improved before it got published in this journal. Also, authors failed to cite the article  (PMID: 30471430) mentioned for adding the table 1. 

There are so many mistakes in putting commas and  (.) in between the manuscript. Check thoroughly the manuscript again and again or I can suggest to use mdpi english editing tool. 

Author Response

February 1, 2023

Dear reviewer:

       We appreciate your thoughtful suggestions and insights. The manuscript has benefited a lot from these insightful suggestions before. The manuscript has been rechecked and the necessary changes have been made in accordance with the reviewer’s suggestions. The responses to the comments have been prepared and attached herewith/given below.

Comments from the Reviewers #1:

      The authors have tried to convince me with the experiments suggested by me. But the manuscript still has flaws. The authors have to rewrite the abstract as it has some mistakes which needs to be improved before it got published in this journal. Also, authors failed to cite the article (PMID:30471430) mentioned for adding the table 1. There are so many mistakes in putting commas and (.) in between the manuscript. Check thoroughly the manuscript again and again or I can suggest to use mdpi english editing tool.

        Response: We apologize for our negligence in citting the article which was drawn on in table 1. The abstract had been rewritten to correct some mistakes. The article (PMID: 30471430) had been cited in line 114. Some mistakes mentioned by reviewers and other grammatical mistake had been also corrected in the manuscript.

Reviewer 2 Report

I appreciate the authors for revising the manuscript as advised

Author Response

February 1, 2023

Dear reviewer:

Comments from the Reviewers #2:

       I appreciate the authors for revising the manuscript as advised.

       We appreciate the reviewer’s invaluable suggestions and thoughtful insights again.